# STING Contributes to Cancer-Induced Bone Pain by Promoting M1 Polarization of Microglia in the Medial Prefrontal Cortex

**DOI:** 10.3390/cancers14215188

**Published:** 2022-10-22

**Authors:** Xiaoxuan Zhang, Xin Li, Wei Wang, Yuxin Zhang, Zhihao Gong, Yuan Peng, Jingxiang Wu, Xingji You

**Affiliations:** 1School of Medicine, Shanghai University, Shanghai 200444, China; 2Department of Anesthesiology, Shanghai Chest Hospital, School of Medicine, Shanghai Jiao Tong University, Shanghai 200030, China

**Keywords:** medial prefrontal cortex, STING, M1 microglia, cancer-induced bone pain

## Abstract

**Simple Summary:**

Cancer-induced bone pain commonly occurs in patients with bone tumors or bone metastases, and in recent years, central nervous system hyperalgesia has received increasing attention. In the present study, we confirmed STING pathways may activate M1 microglia in the mPFC after cancer-induced bone pain. C-176 treatment to inhibit STING can effectively relieve cancer-induced bone pain and pain-related anxiety. In addition, in vivo or in vitro, we both found that C-176 can significantly promote the polarization of M2 microglia and inhibit M1 microglia, which provides a new direction in the treatment of cancer-induced bone pain.

**Abstract:**

The medial prefrontal cortex (mPFC) is the main cortical area for processing both sensory and affective aspects of pain. Recently, mPFC was reported to participate in cancer-induced bone pain (CIBP) via the mechanism of central inflammation. STING is a key component of neuroinflammation in the central neuron system by activating downstream TBK1 and NF-κB signaling pathways. We aimed to investigate whether STING regulated neuroinflammation in the mPFC in rat models of CIBP. It is worth noting that we found a significant upregulation of STING in the mPFC after CIBP, accompanied by activation of TBK1 and NF-κB signaling pathways. In addition, pain and anxiety-like behaviors were alleviated by intraperitoneal injection of the STING inhibitor C-176. Furthermore, in microglia GMI-R1 cells, C-176 reversed LPS-induced M1 polarization. Collectively, this evidence indicated that STING may contribute to cancer-induced bone pain by activating TBK1 and NF-κB, and by promoting M1 polarization of microglia in the mPFC.

## 1. Introduction

Cancer-induced bone pain (CIBP) commonly occurs in patients with bone tumors or bone metastases such as breast, lung, and bladder cancer [1]. Continuous pain seriously affects the patient’s quality of life and causes a variety of syndromes [2]; meanwhile, the causes and pathological mechanisms of cancer-induced bone pain are very complex, including bone destruction caused by tumor invasion, peripheral sensory system hyperalgesia, and central nervous system hyperalgesia [3]. Treatment of CIBP remains a challenge as the drugs used in clinics provide insufficient pain relief; therefore, novel therapeutic targets are urgently needed.

Previous studies have reported that CIBP would alter the synapses and circuits in the medial prefrontal cortex (mPFC), increasing pain sensitivity, and further inducing negative emotions, such as anxiety and depression, which may aggravate the pain and create a vicious circle [4,5,6,7,8]. Neuroinflammation in chronic pain is increasingly being demonstrated by researchers [9,10,11,12,13]: Mao et al. proved that inhibiting neuroinflammation in the spinal cord can relieve CIBP [14]; the study of Wei et al. showed that neuroinflammation of the central spinal may be important for the persistent nociceptive changes in complex regional pain syndrome model [15]. Microglia have been regarded as primary mediators of neuroinflammation in the formation of cancer pain [16,17,18]. Notably, in a peripheral nerve injury model, microglia in mPFC induce neuroinflammation to cause pain [19,20]. Taken together, it can be inferred that the mPFC might be a central hub for chronic pain such as cancer-induced bone pain.

Stimulator of interferon genes (STING) is an endogenous DNA sensor which can be activated together with upstream cGAS by dsDNA of endogenous damaged cells or exogenous pathogens, to participate in pathogen defense, inflammatory response, and tumor immunity [21,22]. Activating STING recruits TANK-binding kinase-1 (TBK1) and phosphorylates IRF3 to relocate to the nucleus where it induces transcription of genes encoding interferons and IFN-stimulated genes such as cytokines and chemokines. Furthermore, TBK1 can lead to activation of the IKK complex and then NF-κB, which translocate into the nucleus providing a synergistic response against invading pathogen [23,24,25,26,27]. According to Donnelly et al., STING modulates type I IFNs to control chronic pain and steady-state nociception [28], while in the central nervous system, STING is generally believed to be involved in neuroinflammation [29,30,31,32]. Microglia is an important immune cell in the central system which plays a vital role in neuroinflammation [33]. In response to microenviromental signals, microglia undergo different types of activation, including: a classic pro-inflammatory (M1) phenotype and an alternative anti-inflammatory (M2) phenotype. Notably, it has been revealed that STING activates M1 microglia to promote inflammation in experimental subarachnoid hemorrhage [34]. Most recently, Sun et al. have shown that in the spinal cord, STING was activated in microglia to induce pain after spared nerve injury [35]. As mentioned above, we speculate that STING in the mPFC might induce neuroinflammation to aggravate CIBP.

In the present study, we hypothesized that in the mPFC, STING might be a crucial regulator in promoting M1 microglia polarization after CIBP. We found a significant upregulation of STING and M1-type microglia polarization in the mPFC after CIBP. Moreover, STING inhibitor C-176 has an antinociceptive effect and reverses the microglia M1 polarization. To our knowledge, the current study is the first to link STING and M1 type microglia polarization in the mPFC to the pathogenesis of CIBP.

## 2. Materials and Methods

### 2.1. Animals and Ethics Statement

Female Sprague–Dawley (SD) rats (200–250 g) and Wistar rats (70–80 g) purchased from B&K Universal Group Limited, were housed 5/cage under a 12 h light-dark cycle at a temperature of 24 ± 1 °C. Over the entire process of the experiment, food and water were accessed ad libitum. All experimental procedures complied with the International Association for the Study of Pain guidelines as used by the Committee of Shanghai Chest Hospital, Shanghai Jiao Tong University.

### 2.2. Cancer-Induced Bone Pain Rat Model

The CIBP model was established according to the previous description by [36]. Briefly, Walker 256 breast cancer cells (purchased from ATCC) were intraperitoneally injected into the female Wister rats. A week later, the ascites were collected and adjusted to the final concentration of 2 × 10^7^ cells/mL in saline. Then, SD rats were anesthetized using Sevoflurane. The upper tibial skin of the right leg was cut for about 1 cm exposing the tibia, the periosteum was scraped off, and holes were drilled. Then, 10 µL Walker 256 cells or normal saline was injected into the bone marrow cavity with a 25 µL syringe, and the hole was sealed with bone wax.

On day 10 after tumor inoculation, vehicle (corn oil, 200 µL) (Sigma-Aldrich, Burlington, MA, USA) or the antagonist of STING (C-176) (purchased from MedChemExpress, Monmouth Junction, NJ, USA) dissolved in 200 µL corn oil with the final concentration of 5.25 µmol [37] were injected intraperitoneally.

### 2.3. Pain Behavior Test

***Paw withdrawal mechanical threshold (PWT).*** As previously reported [36], the von Frey filaments were used to determine the paw withdrawal threshold for paws in response to mechanical stimuli [38]. Briefly, rats were individually placed in a separate compartment with a metal mesh floor. After the rats adapted for about 30 min, von Frey filaments were vertically used to stimulate the right hind paw’s plantar surface for 3 to 5 s. Ascending order of force is used throughout the process (1 g, 1.4 g, 2 g, 4 g, 6 g, 8 g, 10 g, and 15 g), starting at 1 g and ending at 15 g [38]. Paw withdrawal, shaking, and licking were considered to be positive responses. PWT was defined as the minimum force (in grams) required to elicit positive responses to the behavior tests and evaluation performed in a blinded manner.

***Limb use score (LUS).*** As previously reported [36], the rats were placed in an open space, and after balancing for 5 min, the use of limbs during free walking was observed. It is divided into five levels according to the use of hind limbs: 4, normal walking; 3, slight limp of hind limb; 2, aggravated lameness of the hind limb, but still able to be used; 1, severe limping of the hind limb, dragging; 0, hind legs completely off the ground. The whole experiment was conducted by researchers who were blinded to the group settings, behavior tests and evaluation of that performed in a blinded manner.

### 2.4. Elevated Plus Maze Test

Anxiety-like behavior was measured using the elevated plus-maze (Shanghai Jiliang Software Technology Co., Ltd., Shanghai, China) which consists of a central open platform (10 cm × 10 cm), two opposing open arms (50 cm × 10 cm × 0.5 cm), and two opposing closed arms (50 cm × 10 cm × 40 cm). Each rat was placed on the central platform and allowed to explore the apparatus for 5 min while facing one of the open arms. The time spent in the open arms was recorded by JLBehv-EPMG-4 (Shanghai Jiliang Software Technology Co., Ltd., Shanghai, China) based on previous literature [39]; behavior tests and evaluation were performed in a blinded manner.

### 2.5. Bone Histomorphometric Analysis

On day 14 after the model was established, the ipsilateral tibia of CIBP rats was fixed in 4% paraformaldehyde for 48 h, followed by decalcification in 10% EDTA for 3–4 weeks, and 6-μm sections were prepared on a rotating microtome. Paraffin-embedded sections were deparaffinized in xylene, rehydrated, and stained with hematoxylin-eosin (H&E; Sigma, St. Louis, MO, USA) according to the manufacturer’s protocol [40].

### 2.6. Cell Culture

Cells of GMI-R1 (rat microglia) [41] were cultured in H-DMEM with 10% FBS and 1% antibiotics (penicillin and streptomycin, 100 U/mL) at 37 °C in a humidified atmosphere of 5% CO_2_ and passaged every 2–3 day as previously reported [34]. LPS (1 μg/mL) and 20 μM C-176 were administered in the GMI-R1 cells to investigate the role of STING in promoting M1 microglia.

### 2.7. mtROS Measurement

Mitochondrial reactive oxygen species (mtROS) were measured using MitoSOX Red dye according to the manufacturer’s instructions. Following a 24-h treatment with LPS (1 μg/mL), GMI-R1 cells were stained with 5M MitoSOX Red dye for 10 min at 37 °C, washed with PBS, and imaged at 594 nm emission light.

### 2.8. Western Blot Analysis

Western blot analysis was performed as described previously [42], the brain was removed, and the medial prefrontal cortex was sampled and then quickly stored in liquid nitrogen. The medial prefrontal cortex segment was dissected and homogenized in RIPA lysis buffer (150 mM NaCl, 10 mM Tris, 1% Triton X-100, 0.5% NP-40, and 1 mM EDTA at pH 7.4) containing a 1:100 (*v*/*v*) ratio of protease and phosphatase inhibitor cocktail (Roche). The protein concentrations were determined using the bicinchoninic acid (BCA) method. Approximately 50 mg of protein was denatured and separated using 10% SDS PAGE and subsequently transferred to nitrocellulose membranes via electroblotting. After transfer, the membrane was blocked with 5% skimmed milk in a Tris buffer and incubated with specific antibodies: STING (1:1000, 50494, Cell Signaling Tech, Danvers, MA, USA), NF-κB (1:1000, 8242, Cell Signaling Tech, MA, USA), Phospho-NF-κB p65 (Ser536) (1:1000, 3036, Cell Signaling Tech, MA, USA), TBK1 (1:1000, ab40676, Abcam, Cambridge, UK), Phospho-TBK1/NAK (Ser172) (1:1000, 5483, Cell Signaling Tech, MA, USA), MHCII (1:1000, SC-06-78, Invitrogen, Carlsbad, CA, USA), IL-1β (1:500, 515598, Santa cruz, CA, USA), CD206 (1:1000, A8301, Abclonal, Wuhan, China), and GAPDH (1:1000, A19056, Abclonal, Wuhan, China) overnight at 4 °C. After washing, the blots were incubated with horseradish peroxidase-conjugated second antibodies. The quantification of protein expression was normalized to GAPDH using a densitometer (Imaging System). The levels of phospho-NF-κB and phospho-TBK1 were normalized to the unphosphorylated type of these proteins.

### 2.9. Immunofluorescence Staining

Rats were anesthetized by overdose intraperitoneal injection of sodium pentobarbital, then perfused with 100 mL saline and 200 mL 4% paraformaldehyde, and finally, fixed with 4% paraformaldehyde for 72 h. For the removal of residual OCTs, 20 μm sections of the brain were cut and thoroughly rinsed in PBS. The sections were blocked in 0.3% Triton-X 100 and 5% normal donkey serum in PBS for 1 h, then incubated overnight with specific antibodies: STING (1:100, 19851-1-AP, Proteintech, Rosemont, IL, USA), IBA-1 (1:100, ab178847, Abcam, Cambridge, UK), NEUN (1:100, ab177487, Abcam, Cambridge, UK), GFAP (1:100, ab254082, Abcam, Cambridge, UK), and MHCII (1:50, MCA46GA, Bio-rad, Hercules, CA, USA). After incubation and washing with PBS, sections were incubated with secondary antibody solutions (1:500, from Jackson ImmunoResearch, PA, USA or Cell Signaling Technology, MA, USA) at room temperature for 1 h. After rinsing in PBS, the samples were counterstained with DAPI (Beyotime, Nanjing, China) for 3 min. A TCS SP8 fluorescence confocal microscope (Leica Microsystems, Mannheim, Germany) was used to capture and process the images. Images were analyzed using Image-J software (National Institutes of Health, USA).

### 2.10. Data Analysis and Statistics

Data are shown as the mean ± standard error of mean (S.E.M.). Normal distribution and homogeneity of variance, one-way analysis of variance (ANOVA) were used to analyze the differences among the groups. A two-way ANOVA test with repeated measurements was used to analyze the differences in latency over time among groups. The behavior test was repeated three times, and *p* < 0.05 was considered statistically significant. All statistical analyses were performed using GraphPad Prism 8.0 for Windows (San Diego, CA, USA).

## 3. Results

### 3.1. Intratibial Inoculation of Walker 256 Cells Upregulated STING Expression in the mPFC Accompanied by the Hyperalgesia Pain Behavior

Cancer-induced bone pain was induced by intratibial inoculation of Walker 256 cells. The pain behavior tests were conducted on days 0, 4, 7, 11, 14, and 21 (Figure 1A). From day 7 to day 21, the paw withdrawal mechanical threshold (PWT) and limb use score (LUS) significantly decreased in the CIBP group, indicating hyperalgesia (Figure 1B). In addition, the elevated plus-maze experiment was carried out at day 21 and the trajectory paths were analyzed. The test revealed significant differences in the percentage of open arm retention time between sham and CIBP groups (Figure 1C). Results showed that compared with sham rats, CIBP rats spent less time exploring open arms and tended to stay on closed arms, indicating pain-related anxiety. Finally, HE-staining of the tibia also confirmed that at day 21, CIBP rats showed serious bone destruction. According to our findings, the CIBP model was successfully established, accompanied with pain and pain-related anxiety (Figure 1D).

To illustrate the role of STING in mPFC during the development of CIBP, we analyzed the expression of STING in mPFC at days 0, 7, 14, and 21. Western blot analyses showed that the expression of STING increased time-dependently after the CIBP model was established and peaked at day 14 (Figure 1E). Further cellular localization of STING in the mPFC was conducted at day 14 (Figure 1F). Double immunostaining of STING with cell markers (microglial marker:IBA; astrocyte marker:GFAP; neuron marker:NEUN) demonstrated its presences in the mPFC after CIBP. STING was mainly expressed in microglia cells, rather than astrocyte or NEUN (Figure 1F).

The downstream signaling pathways of STING were further verified by Western blot and immunofluorescence at day 14 after CIBP; Western blot and immunofluorescence both indicated that the phosphorylation of TBK1 and NF-κB was significantly increased in the mPFC after CIBP (Figure 2A–D). Accordingly, STING and its downstream signaling pathways might participate in the development of CIBP.

### 3.2. STING Antagonist C-176 Relieved Cancer-Induced Bone Pain and Pain-Related Anxiety

The selective antagonist of STING (C-176) was treated to investigate STING’s potential role in the mPFC after CIBP. Rats were randomly allocated to three groups: sham, CIBP + vehicle (oil corn), and CIBP + C-176. The C-176 or vehicle was intraperitoneally injected on day 10 in CIBP rats. Pain behavior tests were conducted at 0 h, 1 h, 2 h, 4 h, 24 h, 48 h, 72 h, and 96 h after the injection of C-176, the result indicating that C-176 injection can effectively alleviate cancer-induced bone pain from 2 h to 96 h (Figure 3A). Furthermore, in the anxiogenic test, C-176 treatment significantly reduced the percentage of open arm retention time for CIBP rats at 96 h after injection (Figure 3B).

In addition, Western blot analysis showed that the upregulated phosphorylation of TBK1 and NF-κB was almost fully reversed by C-176 (Figure 3C,D), indicating that antagonist C-176 treatment can inhibit the STING signaling pathway in the mPFC of CIBP rats, which may be the reason for the pain and anxiety-like behavior relief, further confirming STING may exert a pain-promoting effect in the mPFC of CIBP rats.

### 3.3. STING Promotes M1 Phenotype Polarization of Microglia in Rats with Cancer-Induced Bone Pain

It is worth noting that in response to microenviromental signals, microglia undergo different types of activation, including a classic pro-inflammatory (M1) phenotype and an alternative anti-inflammatory (M2) phenotype. Having observed that STING was mainly located in the microglia of mPFC (Figure 4A), we further analyzed the phenotypes of microglia polarization in the mPFC after CIBP. We observed that STING was mainly colocalized with MHCII which is the M1 microglia related genes (Figure 4A). Furthermore, immunofluorescence and Western blot analysis both showed that the expression of MHCII was upregulated in the mPFC of CIBP rats, while the increase can be significantly reversed with the treatment of C-176 (Figure 4B–D). In contrast, the expression of CD206, which is the M2a marker, was significantly elevated in the C-176 + CIBP rats (Figure 4D). Moreover, IL-1β as the neuroinflammatory markers was shown to increase in the mPFC of CIBP rats, and this upregulation can be inhibited by C-176 (Figure 4E). Collectively, the above data indicated that STING may contribute to CIBP via promoting microglia polarization to the M1 phenotype, which is a pro-inflammatory phenotype, STING antagonist C-176 may exert analgesic and anti-anxiety effects by inhibiting the polarization of microglia into the M1 pro-inflammatory phenotype and promoting M2 polarization in the mPFC.

In order to further verifying the mechanism, LPS can induce neuroinflammation in microglia both in vivo and in vitro [43], which has been used to treat GMI-R1 (rat microglia cell lines) for 24 h to investigate the association between STING and microglia polarization, Firstly, Mitosox was used to detect the mitochondrial superoxide levels, which may be the cause of STING activation. The results showed that LPS (1 μg/mL) can increase the superoxide levels of mitochondrial in GMI-R1 cell lines (Figure 5A) and activate STING signaling pathways (Figure 5B). Although the level of mitochondrial superoxide and STING was not affected by the treatment of C-176, the increased phosphorylated level of TBK1 and NF-κB, which is the downstream protein of STING, can be significantly reversed by C-176 (Figure 5C), indicating that C-176 treatment in GMI-R1 cell lines can inhibit STING signaling. Importantly, Western blot analysis of MHCII and CD206 showed that C-176 treatment could reverse the increase in MHCII expression while upregulating CD206 expression, indicating that M1 phenotype of microglia in LPS group was activated, and C-176 treatment inhibited the M1 phenotype and promoted polarization into M2 phenotype (Figure 5D). Finally, Western blot also confirmed the increase in IL-1β in the LPS group can be reversed by C-176 treatment (Figure 5E), which further confirmed that STING can mediate the polarization of microglia into the M1 pro-inflammatory phenotype, which may be the cause of CIBP.

## 4. Discussion

The present study demonstrated that STING in the mPFC might be a crucial regulator in promoting microglia polarization to the M1 phenotype after CIBP. We found STING was upregulated in the mPFC after CIBP, accompanied by the increased phosphorylation of the downstream protein, p-TBK1, and p-NF-κB. Meanwhile, STING was mainly located with the activated M1 microglia (presented by MHCII) in the mPFC. After injecting C-176, an inhibitor of STING, pain and anxiety-like behavior induced by CIBP were relieved. On the other hand, C-176 was shown to promote polarization of M2 microglia and inhibit the M1 phenotype both in vivo and in vitro. Taken together, our study suggested that STING may contribute to CIBP by promoting microglia polarization to the M1 phenotype in the mPFC.

Bone cancer causes unbearable pain and anxiety-like behavior in patients, compromising their daily quality of life [44,45]. In the present study, CIBP rats evoke anxiety-like behavior in addition to mechanical hyperalgesia. A well-established method to test anxiety-like behavior is elevated plus-mazes, in which CIBP rats spent significantly less time on the open arms than on the closed arms, suggesting they suffered from pain-related-anxiety. Evidence is accumulating that M1 microglia polarization and overproduction of pro-inflammatory cytokines in the mPFC are linked to the pathogenesis of chronic pain [46,47,48]. Consistently, we observed that CIBP promoted M1 polarization of microglia in the mPFC, which is characterized by elevated expression of MHCII (M1 marker).

Notably, in this present study, we also found an increased expression of STING in the mPFC of CIBP rats. As previously reported, STING emerge as a key mediator of neuroinflammation [49], and most recently, STING has been reported to promote a chronic inflammatory response in low back pain via activation of the NLRP3 inflammasome [50,51]. In addition, Sun et al. also proved that STING may play a vital role in pain initiation via IL-6 in spinal microglia [35]. While, the role of STING in chronic pain still remains controversial, Donnelly et al. [52] showed that systemic administration of DMXAA (STING activator) reduced spontaneous and ongoing pain; Wang verified that activation of STING in the dorsal root ganglion provides an anti-nociceptive effect in cancer-induced bone pain by directly suppressing nociceptor excitability [28]. Our study found that STING was upregulated after CIBP in the microglia of mPFC, and STING inhibitor C-176 could relieve CIBP. This may suggest that STING plays a different role in the central nervous system and peripheral nervous system. In the central nervous system, including spinal cord and mPFC, STING mainly locates in microglia. When STING is activated, it contributes to M1-microglia polarization, promotes neuroinflammation, and plays a role in the process of chronic pain. According to Wang et al., a dose of 10 nmol of C-176 administered at an early stage (day 3 and day 7) did not relieve CIBP [28]. However, in our study, C-176 with a higher dosage of 375 ng (5.25 µmol, 200 µL) [37] was administered to CIBP rats at a mid-stage (day 10) which significantly reduced mechanical allodynia for at least 96 h. Consistent with our results, in a mouse SNI model, administered C-176 at a dose of 40 ng per mouse relieved pain for at least 4 h, no matter whether administered in the early or late stage of chronic pain. In our study, we observed that STING and the downstream protein, pTBK1, and pNF-κB were upregulated in the mPFC of CIBP rats, which was reversed by C-176, further confirming that C-176 provided pain relief by inhibiting STING signaling.

STING activation has been proved to aggravate neuroinflammation by promoting microglial activation and polarizing into the M1 phenotype in experimental subarachnoid hemorrhage [34] and cerebral ischemia-reperfusion injury [53]. In addition, NF-κB, the downstream protein of STING, is involved in the M1 polarization of macrophages and in regulating proinflammatory cytokine production [54,55,56], indicating that STING may be the initial factor of M1 polarization. To further investigate the specific cell types of STING upregulation in the mPFC of CIBP rats, we performed double immunostaining, and of note, we found that STING was mainly colocalized with MHCII, which has been regarded as a surface marker of M1 microglia [57,58], the activation of MHCII and proinflammatory cytokine IL-1β was reversed in the CIBP + C-176 groups. In contrast, the expression of CD206 which is the marker of M2 microglia has been increased after the treatment of C-176. To further assess the role of STING in microglia polarization, C-176 was used in rat microglia cell GMI-R1. As expected, LPS promoted GMI-R1 cells towards M1 phenotype, accompanied by the upregulation of STING. C-176 treatment can decrease the upregulation of MHCII and increase the expression of CD206, indicating that C-176 can inhibit M1 polarization and promote M2 polarization of microglia.

Several limitations should not be ignored in the current study. Donnelly et al. proved STING activator DMXAA may facilitate analgesia, partly by increasing CD8+ T cells in the microenvironment of bone marrow tumor [52], but recent research suggested that another STING agonist, ADU-S100, activated CD8+ T cells at a low dose, while high doses led to the death of the CD8+ T cells [22,59]. Furthermore, STING may affect the polarization states of microglia by regulating other pathways, including NLRP3 inflammasome activation, metabolic reprogramming, or autophagy [29]. In the CIBP, further research is required to understand how STING promotes the polarization of microglia and whether it affects other types of cells, such as immune cells and neurons.

STING is an endogenous DNA sensor which responds to cytosolic DNA in the context of tumor immunity, cellular senescence, and inflammatory diseases. As a result of unstable chromosomes and mitochondrial dysfunction in tumor cells, genomes and mitochondrial DNA can be segregated and released into the cytosol [60]. It is also possible that tumor-induced damage to DRG neurons leads to the release of dsDNA (including nDNA and mtDNA). The mechanism of upregulation and activation of STING in mPFC needs to be further investigated.

## 5. Conclusions

This study demonstrates that STING may be upregulated in the mPFC and may contribute to cancer-induced bone pain in rats, whereas, therapeutic administration using the STING antagonist C-176 inhibits STING-induced central neuroinflammation and M1 type polarization of microglia. Thus, STING may represent a novel target of an anti-inflammatory therapeutic approach for CIBP treatment.

## Figures and Tables

**Figure 1 cancers-14-05188-f001:**
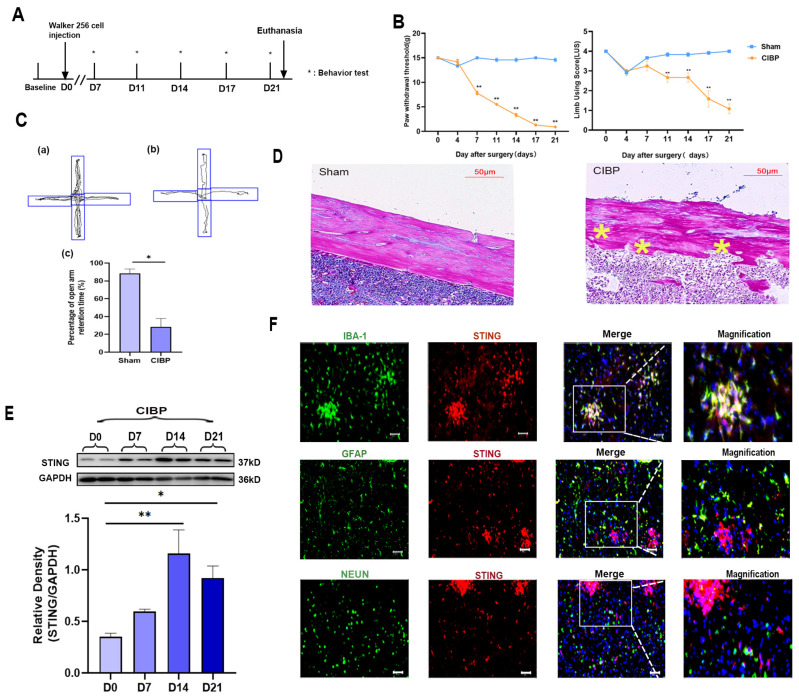
The expression and cellular localization of STING in mPFC of CIBP rats. (**A**) Timeline of the treatment schedule. (**B**) The paw withdrawal threshold (PWT) measured mechanical allodynia (Left). The use of limbs was measured by limb use score (Right), n = 4. (**C**) The tracks and analysis of elevated plus-maze experiment. Representative activity traces of sham (**a**) and CIBP (**b**) in the elevated plus-maze experiment are shown. Percentage of open arm retention time was analyzed (**c**), which was used to measure the degree of anxiety. (**D**) HE staining of tibial on tumor inoculated side. The asterisks are used to mark the areas of bone destruction at day 21 in the CIBP rats. (**E**) The expression of STING at days 0, 7, 14, and 21 was determined using Western blots. GAPDH was detected as an internal control. (**F**) Representative immunofluorescence images of IBA-1 (green), GFAP (green), NEUN (green), and STING (red) in CIBP group rats at day 14. The nuclear was labeled with DAPI (blue), and magnification is shown in the fourth panel. The result showed that STING is mainly located in the microglia of mPFC. Scale bar = 50 μm. Data are expressed as mean ± S.E.M. n = 4, * *p* < 0.05, ** *p* < 0.01 vs. sham group. Full pictures of the Western blots are presented in Appendix A.

**Figure 2 cancers-14-05188-f002:**
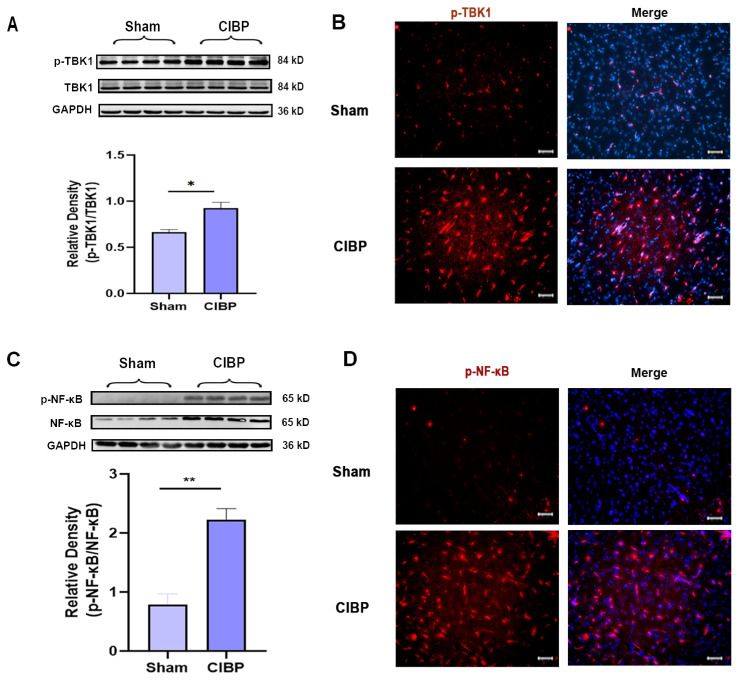
The downstream factors of STING were significantly increased in the mPFC after CIBP. (**A**,**B**) Western blot (**A**) and immunofluorescence (**B**) showed that p-TBK1 increased in the mPFC after CIBP. (**C**,**D**) Western blot (**C**) and immunofluorescence (**D**) showed that p-NF-κB increased in the mPFC after CIBP. The nuclear was labeled with DAPI (blue). GAPDH was detected as an internal control. Data are expressed as mean ±S.E.M., n = 4, * *p* < 0.05, ** *p* < 0.01, CIBP vs. sham. Scale bar = 50 μm. Full pictures of the Western blots are presented in Appendix A.

**Figure 3 cancers-14-05188-f003:**
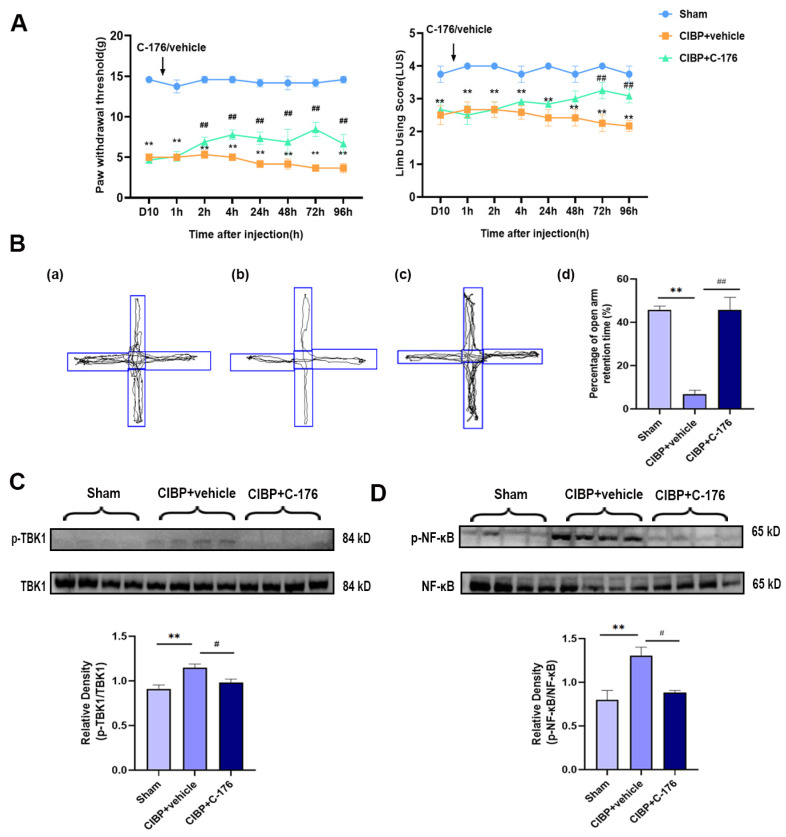
C-176 treatment relieved pain and pain-related anxiety by inhibiting the TBK1-NF-κB signaling pathway in the mPFC of CIBP rats. (**A**) Compared with vehicle (corn oil, 200 µL, i.p.), injection of C-176 (5.25μmol in 200 µL corn oil, i.p.) significantly decreased the level of the paw withdrawal threshold (PWT) (Left) and limb use score (Right) of CIBP rats. (**B**) The tracks and analysis of elevated plus-maze experiment. Representative activity traces of sham (**a**), CIBP + vehicle (**b**), and CIBP + C-176 (**c**) in the elevated plus-maze experiment are shown. (**d**) Percentage of open arm retention time was analyzed’, n = 3. (**C**) The protein expression of the rate of p-TBK1/TBK1 in sham, CIBP + vehicle, and CIBP + C-176 group were determined using Western blots. (**D**) The protein expression of the rate of p-NF-κB/NF-κB in sham, CIBP + vehicle, and CIBP + C-176 group were determined using Western blots. GAPDH was detected as an internal control. Data are expressed as mean ± S.E.M., n = 4. ** *p* < 0.01 vs. sham, ^#^
*p* < 0.05, ^##^
*p* < 0.01 vs. CIBP + vehicle. Full pictures of the Western blots are presented in Appendix A.

**Figure 4 cancers-14-05188-f004:**
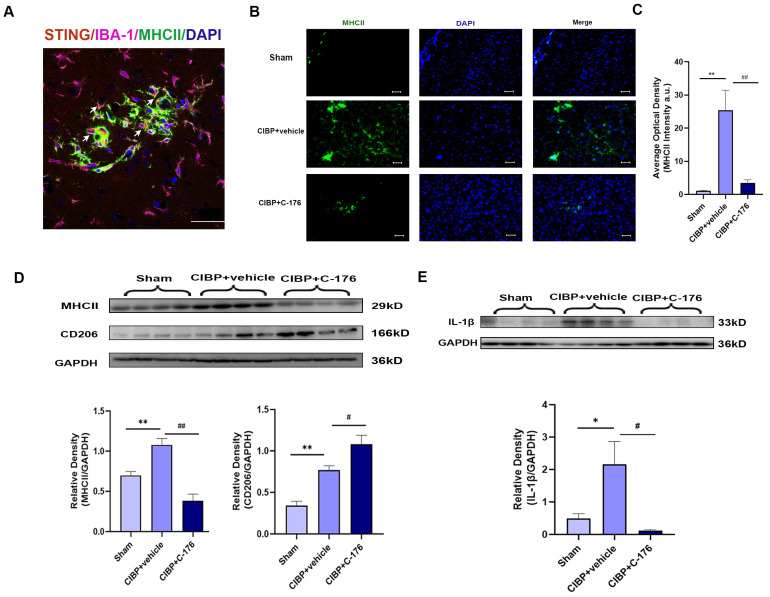
C-176 inhibited the polarization of M1 microglia in mPFC of CIBP model. CIBP rats were i.p. injected with vehicle (corn oil, 200 µL) or C-176 (5.25μmol in 200 µL corn oil) at day 10 after CIBP. (**A**) Immunofluorescence detected the colocalization of STING (red), IBA-1(purple), and MHCII (green) in rats’ mPFC at day 14 after CIBP. (**B**) Representative immunofluorescence images of MHCII (green) in mPFC of sham, CIBP + vehicle, and CIBP + C-176 rats. The nuclear was labeled with DAPI (blue). (**C**) The mean fluorescence intensity of MHCII per image was calculated using Image-J software and normalized to the sham group, n = 3. (**D**,**E**) The protein expression levels in the mPFC of CIBP rats, treated with vehicle or C-176, were assessed using Western blot and quantified: MHCII and CD206 (**D**), IL-1β (**E**) GAPDH was detected as internal control, n = 4. Data are expressed as mean ± S.E.M., * *p* < 0.05, ** *p* < 0.01 vs. sham, # *p* < 0.05, ## *p* < 0.01 vs. CIBP + vehicle. Scale bar = 50 μm. Full pictures of the Western blots are presented in Appendix A.

**Figure 5 cancers-14-05188-f005:**
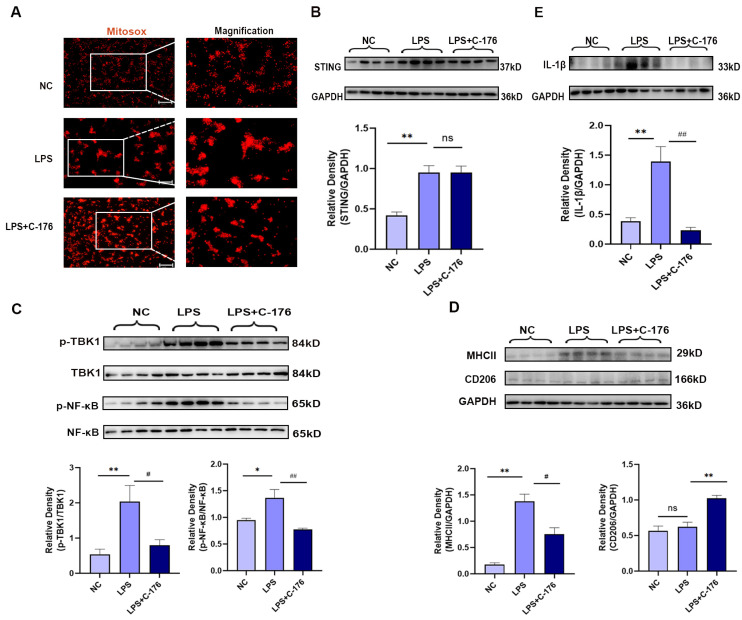
C-176 alleviated LPS-induced polarization of M1 microglia in vitro. (**A**) Representative images of MitoSox-staining in GMI-R1 cells treated with LPS (1μg/mL) or LPS (1 μg/mL) + C-176 (20 μM). The right panel shows magnification. (**B**–**E**) The protein expression levels in GMI-R1 cells, treated with LPS (1 μg/mL) or LPS (1 μg/mL) + C-176 (20 μM), were assessed using Western blot and quantified: STING (**B**); p-TBK1/TBK1 and p-NF-κB/NF-κB (**C**); MHCII and CD206 (**D**); IL-1β (**E**). GAPDH was detected as internal control. NC, negative control. Data are expressed as mean ± S.E.M., n = 4, * *p* < 0.05, ** *p* < 0.01 vs. NC; # *p* < 0.05, ## *p* < 0.01 vs. LPS. Scale bar = 50 μm. Full pictures of the Western blots are presented in Appendix A.

## Data Availability

The data can be shared up on request.

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
