# Peer review of "STING Contributes to Cancer-Induced Bone Pain by Promoting M1 Polarization of Microglia in the Medial Prefrontal Cortex"

_cancers, 2022, doi:10.3390/cancers14215188_

Round 1
Reviewer 1 Report
In this study, authors investigated the role of STING pathway in M1/M2 microglia morphology. They propose that STING contributes to M1 microglia morphology in the medial prefrontal cortex after inoculation of the tumor cell line Walker 256, therefore participating to the neuroinflammatory response in the CNS and alleviation or pain. Overall, the manuscript needs to be improved. Both data and results must be explained more in detail. Please note that the smaller immunofluorescence panels on the left side of figures are useless and quite unreadable. I would consider to remove them, leaving only the larger ones. With regard to what authors state in the "Conclusions", I think the data showed here are not completely sufficient to conclude that "STING may represent a novel target of an anti-inflammatory therapeutic approach...". Perhaps, I would generate data focusing on some neuroinflammatory markers. Additional concerns and comments related to this study are listed below.
Major
· Please note that the list of references is missing in the pdf version of the manuscript. This is fundamental in order to proceed with an accurate review. To me, this is a serious negligence.
· Concerning the Figure 1C, it would be useful to include a detailed description of how the histological sections were made. It worth including arrows to indicate the different tissues/structures showed in the micrographs. Please see “minor” for additional comments about this figure.
· Concerning the western blotting panels illustrated in Figure 2A, I would like to point out that the reactivity of the indicated STING antibody employed in the study is Human and Mouse. Since authors used rat samples, I wonder whether they can prove its specificity in this species, too. The same applies to Phospho-TBK1/NAK and IF panels.
· It would be more informative if the author would have performed the analysis at all time points, from D4 to D14. How is, for example, the trend of STING expression within this period of time?
· Authors suggest that inoculation of tumor cells induces an increase of STING expression (presumably because of the release of DNA from tumor cells). This is supported by the phosphorylation of TBK1 in BCP. However, it has been demonstrated that TBK1 can in turn phosphorylate and induce translocation of STING to subcellular compartments. Do author have data about this mechanism? In addition, does STING activation correlate with citokine production.
· Authors state that, according to IF analysis, STING mainly colocalizes with IBA (microglial marker). However, IBA staining in figure 2 is perplexing. Why there is almost no staining of IBA1 in the sham samples? What about the blu-stained cells in the sham samples of figure 2, upper panels? Based on the staining, they do not seem microglial cells (they are largely negative to IBA1). How authors explain the strong increase of IBA1 expression in BCP samples (uppe panel)? It appears that in BCP there is an increase (proliferation) of micoglial cells. Can authors discuss this results?
· I suppose Figure 2 C and D refers to the western blotting analysis performed at D14. I like to reiterate that it should be more informative to perfom the experiment at each time point. Nevertheless, does NFkB activation by phosphorylation in BCP is truly functional? Can author demonstrate that pNFkB translocate to the nucleus in BCP samples with respect to sham?
· Regarding figure 3, I wonder what is the effect of vehicle and C-176 in the sham sample. I guess this is the correct comparison to do with respect to BCP.
· Line 208. The statement “Having observed that the STING was mainly located in activated microglia..”, must be further experimentally supported.
· Figure 4. These resulta are confusing. It is not clear what authors want to demonstrate. You're supposing that the vehicle (although it is not well specified in the study) is increasing MHCII expression in BCP, therefore promoting the M1 phenoype. What about sham+vehicle and sham+C-178 in these samples?
Minor
· The protocol for histology staining (see Fig. 1 C) is not listed in the Materials and methods. In addition, it is not clear the specific timepoint (D4, 7, 11, 14) at which the histological analysis was performed. This must be indicated in the text, methodology or in the figure caption.
· What about the subcellular distribution of STING? There is no information about this in the manuscript.
· Please note that as they have been submitted, the additional non published data are poorly useful. There is not description or legend that can be of help to undertand their meaning.
· Indicate the source of Walker 256 breast 84 cancer cells
· In immunofluorescence panels of Figure 2 and 4, please correct “megnification”
· Since not all the readers have familiarity with specific cellular markers employed in IF analysis (specifically, IBA and NEUN), authors must describe the rationale of their use in the manuscript.
· Line 77. Please indicate the meaning of SD (rats) abbreviation. Is it Sprague Dawley?
· Line 196. Please, specify who is the “ vehicle” and its meaning.
· Line 209-2011. Classification of microglia phenotypes is (perhaps) more appropriate to the introduction.
· Line 209. Please check the grammar of “Usually, microglia polarized..”
· I would suggest to reorganize the discussion. Perhaps there are parts that sound more suitable to the introduction rather that discussion
Author Response
Response to Reviewer 1 Comments
In this study, authors investigated the role of STING pathway in M1/M2 microglia morphology. They propose that STING contributes to M1 microglia morphology in the medial prefrontal cortex after inoculation of the tumor cell line Walker 256, therefore participating to the neuroinflammatory response in the CNS and alleviation or pain. Overall, the manuscript needs to be improved. Both data and results must be explained more in detail. Please note that the smaller immunofluorescence panels on the left side of figures are useless and quite unreadable. I would consider to remove them, leaving only the larger ones. With regard to what authors state in the "Conclusions", I think the data showed here are not completely sufficient to conclude that "STING may represent a novel target of an anti-inflammatory therapeutic approach...". Perhaps, I would generate data focusing on some neuroinflammatory markers. Additional concerns and comments related to this study are listed below.
Major
- Please note that the list of references is missing in the pdf version of the manuscript. This is fundamental in order to proceed with an accurate review. To me, this is a serious negligence.
Reply:Thank you for your sincerely comments, we have checked the references in the reversed manuscript.
- Concerning the Figure 1C, it would be useful to include a detailed description of how the histological sections were made. It worth including arrows to indicate the different tissues/structures showed in the micrographs. Please see “minor” for additional comments about this figure.
Reply:Thank you for your sincerely comments, we have added the detailed description of how the histological sections were made, and the asterisk was used to markthe destruction area in the CIBP rats.
- Concerning the western blotting panels illustrated in Figure 2A, I would like to point out that the reactivity of the indicated STING antibody employed in the study is Human and Mouse. Since authors used rat samples, I wonder whether they can prove its specificity in this species, too. The same applies to Phospho-TBK1/NAK and IF panels.
Reply:Thank you for your sincerely comments. I checked the antibody of STING, and found a mistake about the STING catalog number. Actually, it was STING (1:1000, 50494, CST), which was positive for reactivity against rat. . Besides, the antibody of Phospho-TBK1/NAK (Ser172) (1:1000, 5483, CST) has reactivity for human and mouse, but in our research, Western blot and immunofluorescence both showed that the expression of p-TBK1 was increased in the mPFC of CIBP rats. While, no immunodetection were observed when horse serum or antibody diluent were substituted for the primary antibodies.
- It would be more informative if the author would have performed the analysis at all time points, from D4 to D14. How is, for example, the trend of STING expression within this period of time?
Reply:Thank you for your sincerely comments, we have checked the expression of STING at D7,D14, and D21. Western blot analyses showed that the expression of STING was increased time-dependently after the CIBP model was established, and peaked at D14 (Figure 1E).
- Authors suggest that inoculation of tumor cells induces an increase of STING expression (presumably because of the release of DNA from tumor cells). This is supported by the phosphorylation of TBK1 in BCP. However, it has been demonstrated that TBK1 can in turn phosphorylate and induce translocation of STING to subcellular compartments. Do author have data about this mechanism? In addition, does STING activation correlate with citokine production.
Reply:Thank you for your sincerely comments. We noticed that TBK1 can in turn phosphorylate and induce translocation of STING to subcellular compartments. In our view, this finding is very interesting and may lead to a better understanding of what activates STING in mPFC of CIBP rats and it gives us a next-step direction. In our study, both STING and IL-1β increased in mPFC of CIBP rats and C-176 reversed the upregulation of IL-1β. Therefore, it is reasonable to believe that STING activation correlates with cytokine production.
- Authors state that, according to IF analysis, STING mainly colocalizes with IBA (microglial marker). However, IBA staining in figure 2 is perplexing. Why there is almost no staining of IBA1 in the sham samples? What about the blu-stained cells in the sham samples of figure 2, upper panels? Based on the staining, they do not seem microglial cells (they are largely negative to IBA1). How authors explain the strong increase of IBA1 expression in BCP samples (uppe panel)? It appears that in BCP there is an increase (proliferation) of micoglial cells. Can authors discuss this results?
Reply:Thank you for your sincerely comments. The blue-stained nuclei in the sham samples of figure 2 might belong to neurons or astrocytes. IBA-1 staining was weak in the sham samples, but strong in the CIBP samples. Research has shown that activated microglia in the brain rapidly proliferate and aggregate and contribute to neuroinflammation after cerebral ischaemia or injury to the CNS. [Li T., Zhang S. Microgliosis in the injured brain: infiltrating cells and reactive microglia both play a role. The Neuroscientist. 2016;22(2):165–170.] In this present study, the strong increase of IBA1 expression in CIBP samples may indicate that microglia are proliferating, migrating, and aggregating. Moreover, our slice showed highlights of IBA staining and microglia aggregation in the mPFC of CIBP. It is reasonable to believe that microglia in the mPFC of CIBP rat rapidly proliferate and aggregate.
- I suppose Figure 2 C and D refers to the western blotting analysis performed at D14. I like to reiterate that it should be more informative to perfom the experiment at each time point. Nevertheless, does NFkB activation by phosphorylation in BCP is truly functional? Can author demonstrate that pNFkB translocate to the nucleus in BCP samples with respect to sham?
Reply:Thank you for your sincerely comments, we detected the expression of STING at different time. Western blot analyses showed that the expression of STING was increased time-dependently after the CIBP model was established, and peaked at D14. As such, we further investigate the mechanism in mPFC at D14.10
According to our immunofluorescence and Western blot results, the level of pNF-kB in BCP group was significantly higher than that in Sham group. Our experimental conditions, however, prevented us from observing whether pNF-kB translocated into the nucleus. In a number of literatures, it is reported that NF-kB can be phosphorylated and translocate from the cytoplasm into the nucleus to exert its function.
410· Regarding figure 3, I wonder what is the effect of vehicle and C-176 in the sham sample. I guess this is the correct comparison to do with respect to BCP.
Reply:I agree with you that the effect of vehicle and C-176 in the sham sample was the reasonable comparison to do with respect to BCP. We regret that we did not set up these two groups. While, our results show that there is no significant difference between the two groups of BCP and BCP+Vehicle(corn oil) in terms of PWT, LUS, and STING expression. To some extent, it can be explained that Vehicle (corn oil) itself does not affect the pain and STING expression of CIBP rats.
- Line 208. The statement “Having observed that the STING was mainly located in activated microglia..”, must be further experimentally supported.
Reply:Thank you for your sincerely comments, we observed that STING was mainly co-localized with IBA-1 which is the marker of microglia. In our experiment, MHCII, a marker of M1 phenotype, was found to co-localize with STING. I agree with you that the statement need further experiment about more M1 microglia markers. I rewrite the sentence as “the STING was mainly located in microglia.”
- Figure 4. These resulta are confusing. It is not clear what authors want to demonstrate. You're supposing that the vehicle (although it is not well specified in the study) is increasing MHCII expression in BCP, therefore promoting the M1 phenoype. What about sham+vehicle and sham+C-176in these samples?
Reply:Thank you for your sincerely comments, in this present study, Western blot and immunofluorescence both showed that MHCII, which is the marker of pro-inflammatory phenotype microglia upregulated in the mPFC of CIBP rats, and this increase can be reversed by inhibiting the STING pathway. Thus, we supposed that STING might induce the activation of M1 microglia to participate in the process of CIBP. While we do not test the expression of MHCII in the Sham+Vehicle and Sham+C-176 rats.
Minor
- The protocol for histology staining (see Fig. 1 C) is not listed in the Materials and methods. In addition, it is not clear the specific timepoint (D4, 7, 11, 14) at which the histological analysis was performed. This must be indicated in the text, methodology or in the figure caption.
Reply:Thank you for your sincerely comments, the protocol for histology staining has been added to the Materials and methods section and the figure caption states that the histological analysis was performed at D14
- What about the subcellular distribution of STING? There is no information about this in the manuscript.
Reply:Thank you for your sincerely comments, we have conducted double immunofluorescence in the mPFC, and our result showed that STING mainly located in microglia of mPFC. The experiment to determine the subcellular distribution of STING requires some more time.
- Please note that as they have been submitted, the additional non published data are poorly useful. There is not description or legend that can be of help to undertand their meaning.
- Indicate the source of Walker 256 breast 84 cancer cells
Reply:Thank you for your sincerely comments, we have added the source of Walker 256 breast cancer cells.
- In immunofluorescence panels of Figure 2 and 4, please correct “megnification”
Reply:Thank you for your sincerely comments, It has been corrected.
- Since not all the readers have familiarity with specific cellular markers employed in IF analysis (specifically, IBA and NEUN), authors must describe the rationale of their use in the manuscript.
Reply:Thank you for your sincerely comments, we have made some detail description of the analysis in the manuscript.
- Line 77. Please indicate the meaning of SD (rats) abbreviation. Is it Sprague Dawley?
Reply:Thank you for your sincerely comments, SD (rats) is Sprague Dawley. We added the details in the latest manuscript.
- Line 196. Please, specify who is the “ vehicle” and its meaning.
Reply:Thank you for your sincerely comments, vehicle we used in this research is corn oil and we added the detail in the manuscript.
- Line 209-2011. Classification of microglia phenotypes is (perhaps) more appropriate to the introduction.
Reply:Thank you for your sincerely comments, we have reorganized the discussion in the latest manuscript.
- Line 209. Please check the grammar of “Usually, microglia polarized..”
Reply:Thank you for your sincerely comments, we have rewritten the sentence and changed it to “In response to microenviromental signals, microglia undergo different types of activation”.
- I would suggest to reorganize the discussion. Perhaps there are parts that sound more suitable to the introduction rather that discussion
Reply:Thank you for your sincerely comments, we have reorganize the discussion in the latest manuscript.

Reviewer 2 Report
In this study, Zhang et al investigated the effects of cancer-induced bone pain (CIBP) on the central nervous system and showed that CIBP increases the expression of STING in the medial prefrontal cortex (mPFC), which in turn activates TBK1 and NF-κB and promotes M1 polarization of microglia. Importantly, the STING inhibitor C-176 reduced CIBP in conjunction with decreased activation of TBK1 and NF-kB and M1 polarization of microglia. From these results, authors propose that STING is a potential novel therapeutic target for CIBP.
Comments
1. The study is important and relevant, and the manuscript overall reads alright, although the text needs minor English editing here and there (e. g. line 314 and 319).
2. The graphical abstract properly shows the summary of the study. However, the mechanism of up-regulation and activation of STING in mPFC by CIBP is not discussed in the figure. For example, how is the peripheral CIBP signal transduced to mPFC? Is that mediated via dorsal root ganglia (DRGs)? Are STING expression and activation in DRGs changed by CIBP? Please include these pieces of information in the graphical abstract and also discuss in Discussion in the text.
3. As shown in Figure 1C, it appears that bone destruction associated with tumor growth in the bone marrow has significant effects on STING expression, activation of TBK1 and NF-κB, and M1 microglia polarization. Have authors had a chance to determine the effects of inhibition of bone destruction by bisphosphonates or denosumab on STING, TBK1 and NF-kB and microglia polarization? If not, please discuss this point in the text.
4. Figure 1C likely shows better at a higher magnification. Please present high power HE pictures in Figure 1C.
5. Were behavior tests and evaluation of that performed in blind manner? If so, please state that in the text.
6. Readers may not be familiar with the microglia cell line GMI-R1 and the STING antagonist C-176. Please provide detailed scientific information of GMI-R1 and C-176 by citing literature, particularly, the specificity of C-176 for STING.
7. In Discussion, authors describe that C-176 was tested at various doses and given at different timing. Considering these previous studies, how did authors choose the administration timing and dose in this study?
8. The points authors state in the text are hard to see in Figure 2B, 4A, and 5A and 5B. Please expand the explanation of these figures so that the points authors want to show readers become much clearer.
9. Please provide literature that shows that LPS treatment of GMI-R1 cells in vitro mimic neuroinflammation.
10. What is BCP in Figures? Please change it to CIBP for consistency.
11. “Megnification” in Figure 2B and 4A should read “magnification”.
12. Line 193 and 196, CLBP should read CIBP.
Author Response
- The study is important and relevant, and the manuscript overall reads alright, although the text needs minor English editing here and there (e. g. line 314 and 319).
Reply:Thank you for your sincerely comments, we have tried our best to correct grammar mistakes in our revised manuscript.
- The graphical abstract properly shows the summary of the study. However, the mechanism of up-regulation and activation of STING in mPFC by CIBP is not discussed in the figure. For example, how is the peripheral CIBP signal transduced to mPFC? Is that mediated via dorsal root ganglia (DRGs)? Are STING expression and activation in DRGs changed by CIBP? Please include these pieces of information in the graphical abstract and also discuss in Discussion in the text.
Reply:Thank you for your sincerely comments. I think it's very interesting to learn the mechanism of up-regulation and activation of STING in mPFC of CIBP. Some possibilities are listed below: dsDNA from tumor cells, dsDNA from damaged neurons, or from dorsal root ganglia. It has been shown that tumor cells can release dsDNA, which can pass the blood brain barrier via exosomes. In the SNI model, dsDNA in the peripheral blood is significantly increased, which may cause STING to be activated by exogenous dsDNA. Actually we have focused dorsal root ganglia in another research, the expression of STING and its signaling pathway has been confirmed upregulated at CIBP rats. While, whether the peripheral CIBP signal transduced to mPFC is still not been explored. In CIBP model, further studies are needed to determine whether STING was activated by dsDNA from tumor cells.
- As shown in Figure 1C, it appears that bone destruction associated with tumor growth in the bone marrow has significant effects on STING expression, activation of TBK1 and NF-κB, and M1 microglia polarization. Have authors had a chance to determine the effects of inhibition of bone destruction by bisphosphonates or denosumab on STING, TBK1 and NF-kB and microglia polarization? If not, please discuss this point in the text.
Reply:Thank you for your sincerely comments, in this research, bone destruction may be the cause of bone pain, it is notably to explore whether the bone destruction associated with the expression of STING, this may be the key point which we can investigate in the future.
- Figure 1C likely shows better at a higher magnification. Please present high power HE pictures in Figure 1C.
Reply:Thank you for your sincerely comments Figure 1C has been replaced with a figure at a higher magnification image and the destruction area has been circled.
- Were behavior tests and evaluation of that performed in blind manner? If so, please state that in the text.
Reply:Thank you for your sincerely comments, behavior tests and evaluation of that performed in blind manner, we have stated that in the section of “Materials and Methods”.
- Readers may not be familiar with the microglia cell line GMI-R1 and the STING antagonist C-176. Please provide detailed scientific information of GMI-R1 and C-176 by citing literature, particularly, the specificity of C-176 for STING.
Reply:Thank you for your sincerely comments, we have provide the literature on GMI-R1 and C-176 in the latest manuscript.
- In Discussion, authors describe that C-176 was tested at various doses and given at different timing. Considering these previous studies, how did authors choose the administration timing and dose in this study?
Reply:Thank you for your sincerely comments. Considering that cancer-induced bone pain always occurs in the middle or late stages, we chose D10 for administration, which corresponded to the middle and late stages of CIBP in rats. Besides, Haag, who screened and synthesized the small molecule compound C-176, recommended a dosage of 750 nmol almost 53 ng per mice. Based on the body surface area, C-176 with a dose of 375 ng (5.25umol, 200ul) was administered to CIBP rats and significantly reduced mechanical allodynia for at least 96 hours.
- The points authors state in the text are hard to see in Figure 2B, 4A, and 5A and 5B. Please expand the explanation of these figures so that the points authors want to show readers become much clearer.
Reply:Thank you for your sincerely comments, we checked figure, and added the detail information about this figures.
- Please provide literature that shows that LPS treatment of GMI-R1 cells in vitro mimic neuroinflammation.
Reply:Thank you for your sincerely comments, we have provide the literature on LPS treatmentof murine microglia cell line BV2 and primary microglia to mimic neuroinflammation [Yang L, Zhou R, Tong Y, et al. Neuroprotection by dihydrotestosterone in LPS-induced neuroinflammation. Neurobiol Dis. 2020 Jul;140:104814.]. In this present research, we evidenced that LPS increased the expression of proinflammatory factor IL-1β. Therefore, the microglia cell line GMI-R1 is a reasonable model for mimicking neuroinflammtion.
- What is BCP in Figures? Please change it to CIBP for consistency.
Reply:Thank you for your sincerely comments, we have changed BCP to CIBP
- “Megnification” in Figure 2B and 4A should read “magnification”.
Reply:Thank you for your sincerely comments, we have now corrected the spelling errors.
- Line 193 and 196, CLBP should read CIBP.
Reply:Thank you for your sincerely comments, we have now corrected the spelling errors.

Round 2
Reviewer 1 Report
The overall quality of manuscript has slightly improved. In my opinion, there are still some concerns that must be solved to make the manuscript suitable for publication in Cancers. In particular, the quality and the readability of figures. As a general suggestion, all panels included in the figures should be clearly identified and listed/discussed both in the text and figure captions. Please note that not all the panels are clearly explained in the text and captions. This is the case of densitometric analyses for example. Notably, most immunofluorescence panels are still hard to be read. I would suggest 1) to indicate each panel by a specific letter 2) to list/illustrate the meaning of each panel in the text/caption, 3) to remove any confusing graphical symbols, forms, etc. Figures 1, 4 and 5 show some white squares in the immunofluorescence images that are not explained anywhere. What’s the meaning of these boxes? The reader should not be forced to "grasp" the meaning of each single panel. What the fourth column of figure 1 and 4 are showing? In addition, the scale bars (and sometimes the text) in IF images are unreadable. The magnification of IF panels (for example in fig 4) appear different if one compares the first three columns with the fourth. This is not acceptable. I strongly suggest review more carefully the overall quality of data and figures before submitting the manuscript. As it is, in my opinion, the manuscript is still not acceptable for publication.
Author Response
Author reply
Thank you for your sincerely suggest. Firstly, for the figure captions, we rewrote the figure legend in the manuscript to illustrate the meaning of each panel. Secondly, in the figures especially figures 1, 4 and 5, we have thickened the scale bars, and marked the scale bar=50mm in the caption. Furthermore, the white square in the figure is where we selected a typical location and enlarged it in the next right-hand panel. As your suggestion, all panels included in the figures are clearly identified and discussed both in the text and figure captions.
Reviewer 2 Report
The manuscript has been reasonably revised in response to the comments of this reviewer.
Why has graphical abstract been deleted in the revised manuscript? It comprehensively shows the entire study for readers. Please show it again in the manuscript by including some possibilities the authors assume.
Author Response
Thank you for your sincere comment, the graphical abstract was uploaded in a separate file.
And we have reversed the manuscript now with the graphical abstract.
Round 3
Reviewer 1 Report
Thank you for further revising your manuscript. Authors have fulfilled the previous requests making IF panels at least more readable now and acceptable for publishing (although they are still not of best quality).
Author Response
Thank you for reviewing. We will strive to improve the quality of the manuscript.